# Evaluating the Natural History of Groin Hernia from an “Unplanned” Watchful Waiting Strategy

**DOI:** 10.3390/jcm12124127

**Published:** 2023-06-19

**Authors:** Marco Ceresoli, Stella Konadu Adjei Antwi, Megi Mehmeti, Serena Marmaggi, Marco Braga, Luca Nespoli

**Affiliations:** General and Emergency Surgery, School of Medicine and Surgery, Milano-Bicocca University, Fondazione IRCCS San Gerardo dei Tintori, Via Pergolesi 33, 20900 Monza, Italy; s.adjei@campus.unimib.it (S.K.A.A.); m.mehmeti@campus.unimib.it (M.M.);

**Keywords:** groin hernia, watchful waiting, conservative management, pandemic, emergency hernia surgery

## Abstract

Groin hernia is one of the most common surgical diagnoses worldwide. The indication for surgery in asymptomatic or mildly symptomatic patients is discussed. Some trials have demonstrated the safety of a watchful waiting strategy. During the pandemic, waiting lists for hernia surgery dramatically increased the opportunity to evaluate the natural history of groin hernias. The present study aimed to evaluate the incidence of emergency hernia surgery in a large cohort of patients that were selected and were waiting for elective surgery. This is a retrospective cross-sectional cohort study including all patients evaluated and selected for elective groin hernia surgery at San Gerardo Hospital between 2017 and 2020. Elective and emergency hernia surgeries were recorded for all patients. The incidence of adverse events was also evaluated. Overall, 1423 patients were evaluated, and 964 selected patients (80.3%) underwent elective hernia surgery, while 17 patients (1.4%) required an emergency operation while waiting for an elective operation. A total of 220 (18.3%) patients were still awaiting surgery in March 2022. The overall cumulative risk levels for emergency hernia surgeries were 1%, 2%, 3.2%, and 5% at 12, 24, 36, and 48 months, respectively. There was no association between longer waiting periods and an increased need for emergency surgery. Our study indicates that up to 5% of patients with groin hernia require emergency surgery at 48 months from the evaluation; the increased waiting time for surgery for elective groin hernia repair was not associated with an increased incidence of adverse events.

## 1. Introduction

Groin hernia is one of the most common surgical pathologies worldwide. Groin hernias are a frequent cause of pain symptoms in adulthood; from patients presenting as asymptomatic to those presenting with severe complications (e.g., bowel obstruction or incarceration). Each year, approximately 20,000,000 patients undergo inguinal hernia repair worldwide, with this figure increasing annually [1].

One of the principal indications for asymptomatic or mild symptomatic hernia surgery is the prevention of hernia incarceration and strangulation with the need of emergency surgery. However, the lifetime risk of inguinal hernia strangulation has been estimated to be between 0.27 and 0.03% [2]. Three randomized controlled trials demonstrated that a watchful waiting strategy is safe for asymptomatic or minimally symptomatic inguinal hernias [3,4,5]. A pre-pandemic study found that for male patients with mildly symptomatic inguinal hernias who were followed up non-operatively (i.e., watchful waiting), the long-term chance of requiring emergency hernia repair was 0.2% [3]. Barry de Goede et al. [4] compared a watchful waiting approach to elective hernia repair in men aged 50 and older with mild or asymptomatic inguinal hernia. Notably, about 2.3% of the conservatively treated patients required emergency surgery for strangulation/incarceration.

Patient candidates for elective hernia repair are required to be placed on a waiting list, and a relatively long waiting period could pass between the initial evaluation and subsequent surgical procedure. Several factors can affect this time interval, such as the high volume of patients and the restricted resources and availability. Recently, the outbreak of the COVID-19 pandemic dramatically affected healthcare systems, delaying many non-urgent services [6,7,8]. While efforts were made to partially preserve oncological activities, patients selected for non-oncologic surgical procedures, such as groin hernia repair, were forced to endure a longer waiting period. From a certain perspective, this longer waiting time can be considered an unplanned watchful waiting approach. In this regard, a longer waiting time could have affected the incidence of adverse events related to the hernia (e.g., strangulation and incarceration). Therefore, this unusual scenario provided us with the possibility to evaluate the natural history of groin hernias and the need for emergency surgery related to hernia complications.

The present study aimed to evaluate the natural history of groin hernias in a non-selected cohort of patients awaiting elective surgical repair.

## 2. Materials and Methods

This was a retrospective cross-sectional study including all patients evaluated in an outpatient service (symptomatic or asymptomatic) for inguinal or femoral hernias and selected for elective surgical repair from 1 January 2017 to 31 December 2020 at San Gerardo Hospital, Monza, Italy. Follow-up and patients’ conditions were assessed in March 2022. The elective or emergency hernia surgery performed was recorded for all patients. In March 2022, all patients who remained present on our surgical waiting list were contacted via phone by three investigators to complete the follow-up, asking about potential surgical hernia repair in other hospitals (elective or unplanned). In the case of planned or unplanned surgery, the reason for surgery and details about the surgical intervention along with postoperative complications were collected. The EuraHS QoL questionnaire [9] was administered to all patients at the time of the follow-up.

For each patient, their age, sex, working activity, hernia site (inguinal or femoral), and the time interval between the first evaluation and surgery were collected.

The study population was divided into two cohorts of patients based on the period of evaluation. The study time frame was divided into two periods to evaluate the effect of the pandemic. Given the median waiting time of 10 months, we identified two study groups: patients evaluated in 2017–2018 and patients evaluated in 2019–2020. The risk of emergency hernia surgery was evaluated and compared between the two groups and between patients with inguinal and femoral hernias.

Continuous data are shown as median and interquartile ranges, while categorical data are shown as percentages. The incidence of unplanned surgery was evaluated using the Kaplan–Meier method and compared with the log-rank test. Hazard rates for emergency hernia surgery were calculated using univariate and multiple Cox regression methods. Paired continuous data were analyzed using the paired *t*-test.

Analysis was performed using SPSS v 28 (IBM Corp. Released 2021. IBM SPSS Statistics for Windows, Version 28.0. Armonk, NY, USA: IBM Corp).

## 3. Results

During the study period, a total of 1759 groin hernia procedures were performed at our institution. The number of elective procedures per year varied significantly due to the pandemic, decreasing from 383 in 2018 to 162 in 2020–2021. Time to surgery also increased for the same reason; the median waiting time was 8.83 months (95%CI 8.21–9.45) for patients evaluated in 2017–2018 and 17.63 months (95%CI 13.48–21.81) for patients evaluated in 2019–2020 (*p* < 0.001) (Figure 1 and Figure 2).

Overall, 1425 patients with inguinal or femoral hernias were evaluated in an outpatient setting during the study period and selected for elective surgery. Overall, 222 patients were lost at follow-up and thus were not included in the analysis. Upon evaluating both inguinal and femoral hernias, a total of 964 (80.3%) patients underwent elective surgery, while 17 patients (1.4%) required an emergency operation, the majority with femoral hernias. A total number of 220 (18.3%) patients were still awaiting surgery in March 2022.

### 3.1. Inguinal Hernia

In 2017–2018, 686 patients with inguinal hernia were selected for surgery, while 492 patients were selected for surgery in 2019–2020 (29% decrease). Patients’ characteristics were similar between the two groups. Table 1 presents patients’ characteristics in detail.

In the 2017–2018 cohort, 625 (91.1%) patients underwent elective surgery, while 54 (7.9%) patients were still waiting. In the 2019–2020 cohort, 322 (65.4%) patients underwent elective surgery, while 163 (33.1%) were still waiting. No significant difference was observed between the two cohorts in terms of emergent surgical procedures: seven patients (1.1%) in the 2017–2018 cohort underwent emergency surgery compared to six patients (1.2%) in the 2019–2020 cohort.

The overall cumulative risks for emergency hernia surgeries were 1%, 2%, 3.2%, and 5% at 12, 24, 36, and 48 months, respectively. Figure 3 presents the cumulative risk of emergency hernia surgery between the two study cohorts: there were no differences (*p* = 0.884) with similar risks. In the univariate analysis (Table 2), no factors were associated with the need for an emergency procedure.

Among the thirteen patients who underwent emergency hernia surgery, two required laparotomies with bowel resection, and one died of postoperative intra-abdominal severe sepsis due to an anastomotic leak.

After a median waiting period of 29.3 months (IQR 22.5–40.0), the remaining 220 (18.3%) patients on the waiting list had no observed adverse effects at the time of evaluation. Table 3 shows that no significant variations in EuraHS QoL scores were observed between the first evaluation and the last follow-up.

### 3.2. Femoral Hernia

During the 2017–2018 time interval, 16 patients were visited in an outpatient service and selected for surgery: 12 patients (75%) had elective surgery, and 1 (6.25%) patient underwent an emergency procedure. In the 2019–2020 cohort, seven patients were selected for elective surgery, four of them (57%) were operated on in an elective setting, and three (43%) patients required an emergency procedure. Patients’ characteristics are presented in Table 1.

The overall risks for emergency surgeries were 10 and 46% at 12 and 36 months, respectively.

## 4. Discussion

The present study has shown that the incidence of unplanned emergency hernia surgery in a large cohort of patients with groin hernia was approximately 1.4% after a median follow-up of 10 months (IQR 4–21), with a strangulation risk of 0.25%. A two-fold increase in waiting time for elective surgery did not result in a higher number of unplanned surgeries or greater strangulation risk over time.

Groin hernia is a very common situation, and hernia repair is one of the most common surgical operations performed worldwide every year. Hernia surgery is motivated by the presence of symptoms and the discomfort of patients, which cause limitations in daily activities and an impact on quality of life, despite the majority of patients having no pain or being mildly symptomatic at the time of surgery [10]. Another important reason for surgical correction is the prevention of hernia incarceration with the need for emergency surgical hernia repair. On the other hand, hernia surgery is also associated with some disadvantages, such as hernia recurrence and chronic pain that could affect up to 4 and 12% of operated patients, respectively [11,12].

The natural history of an untreated groin hernia remains a debated topic. Three randomized controlled trials have demonstrated that a watchful waiting strategy for asymptomatic or minimally symptomatic inguinal hernia is a safe treatment option [3,4,5]. A recent systematic review and meta-analysis confirmed this result, reporting an estimated emergency surgical hernia repair rate of 2–3% [13]. Other studies estimated the lifetime strangulation risk of inguinal hernias to be between 0.27 and 0.03% [2].

The impact of the COVID-19 pandemic on elective non-oncologic surgery resulted in a decrease in surgical volume and an increase in time spent on a waiting list, as confirmed by our data. The prolonged unplanned waiting time for elective surgery could be considered similar to a watchful waiting strategy. In light of this analogy, our results also confirmed the safety of this strategy in an unselected cohort of patients, reflecting the daily clinical practice.

The need for emergency surgical hernia repair due to incarceration or strangulation is an undesirable event. Different from elective hernia surgery, where morbidity and mortality are minimal, emergency hernia surgery is burdened by significant morbidity and mortality rates [14]. In our cohort of 17 patients who underwent emergency surgery, 4 (23.5%) patients needed an explorative laparotomy and 3 (17.6%) required a bowel resection. Overall, we recorded one case of mortality (5.9%) and some minor morbidities (11.7%). In a previous study including 259 elderly patients submitted to emergency hernia surgery, the mortality and morbidity rates were 2.8 and 21.2%, respectively. Factors related to morbidity and mortality included the need for laparotomy and bowel resection, higher comorbidities, and altered mental status [15]. Another study reported that emergency hernia repair in the elderly was burdened by high morbidity (33%) and high mortality when compared with elective surgery [16].

In a complex scenario such as a pandemic, the indispensable shift of hospital resources yielded a reduction in elective non-oncologic surgeries, including hernia repair. However, our data showed that this reduction was not associated with a significant increase in major adverse clinical events.

Despite its relevant insights, the main limitation of our study is its retrospective design. Moreover, due to the low incidence of emergent events related to hernia incarceration, the study could be underpowered in terms of detecting small differences. However, the present study has the strength of reporting data derived from daily practice and not from a highly selected cohort of patients, reinforcing the clinical significance of the results.

In conclusion, our study shows that an increased waiting time for elective inguinal hernia repair surgery was not associated with an increased incidence of adverse events in terms of emergent repair; with a waiting time of more than double, the incidence of emergency surgical repair did not change. Analogous results were not observed concerning the femoral hernia.

## Figures and Tables

**Figure 1 jcm-12-04127-f001:**
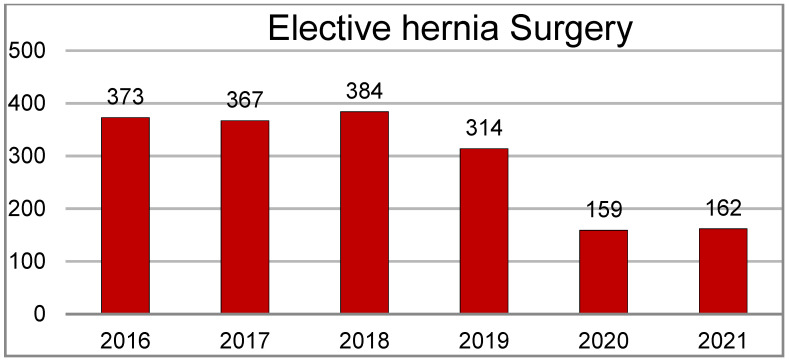
Elective groin hernia surgery volumes at San Gerardo Hospital.

**Figure 2 jcm-12-04127-f002:**
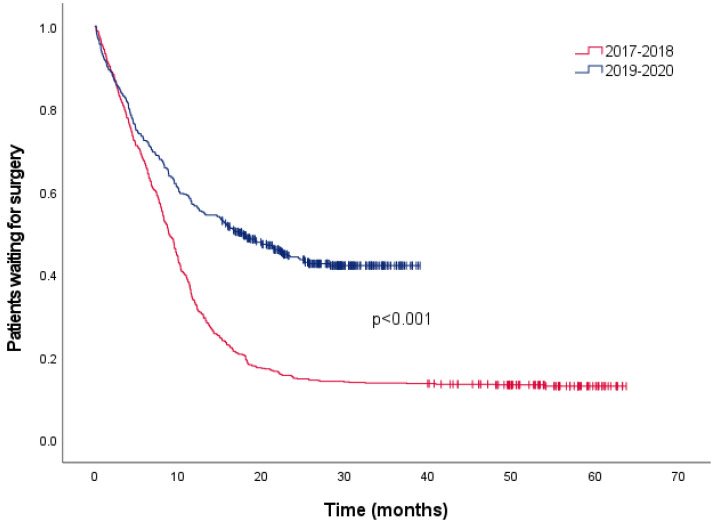
Time to elective surgery at San Gerardo Hospital.

**Figure 3 jcm-12-04127-f003:**
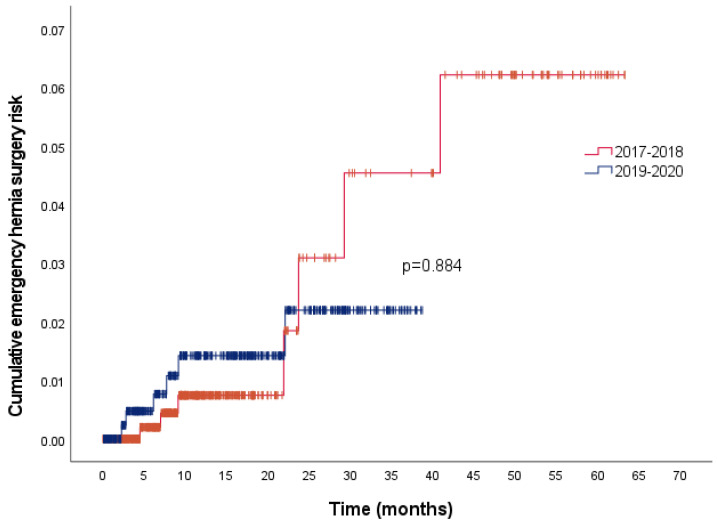
Emergency hernia surgery risk in patients with inguinal hernia.

**Table 1 jcm-12-04127-t001:** Patients’ characteristics.

		Inguinal Hernia					Femoral Hernia				
		2017–2018		2019–2020			2017–2018		2019–2020		
		N/median	%/(IQR)	N/median	%/(IQR)	*p*-Value	N/median	%/(IQR)	N/median	%/(IQR)	*p*-Value
Number of patients		686		492			18		7		
Sex	F	53	7.7%	46	9.3%	0.32	12	75.0%	7	100.0%	0.12
	M	633	92.3%	446	90.7%		4	25.0%	0	0.0%	
Age		66.97	(54.71–74.76)	66.53	(56.68–76.84)	0.76	56.43	(47.70–71.87)	67.37	(47.88–73.55)	0.09
Age > 70		278	41.8%	205	42.4%	0.85	5	31.3%	2	28.6%	0.96
Occupation	Retired	386	56.2%	286	58.2%	0.583	7	44.0%	4	57.10%	0.253
	Sedentary	158	23.1%	106	21.5%		2	12.50%	1	14.30%	
	Moderate manual	33	4.8%	29	6%		3	18.50%	1	14.30%	
	Heavy manual	109	15.9%	71	14.3%		4	25.0%	1	14.30%	
Number of patients per Year	2017	374	54.5%	0	0.0%		13	81.3%	0	0.0%	
	2018	312	45.5%	0	0.0%		3	18.8%	0	0.0%	
	2019	0	0.0%	278	56.5%		0	0.0%	4	57.1%	
	2020	0	0.0%	214	43.5%		0	0.0%	3	42.9%	
Surgery		632	92.1%	328	66.7%	<0.001	13	81.3%	7	100.0%	0.55
Emergency surgery		7	1.0%	6	1.2%	0.75	1	6.3%	3	42.9%	0.25
Laparotomy and bowel resection		1	0.01%	1		0.98	0	0.0%	0	0.0%	0.98
Mortality		1	0.01%	0	0.0%	0.68	0	0.0%	0	0.0%	0.97
Status	Still in Waiting list	54	7.9%	163	33.1%	<0.001	3	18.8%	0	0.0%	0.87
	Operated (our center)	597	87.0%	271	55.1%		14	87.5%	7	100.0%	
	Operated (other center)	35	5.1%	58	11.8%		0	0.0%	0	0.0%	

Legend: IQR = interquartile range; N = number.

**Table 2 jcm-12-04127-t002:** Univariate analysis of emergency surgery risk in inguinal hernia patients.

		Hazard Ratio	95% CI		Sign.
			Inferior	Superior	
Sex	Female	1 (ref)	-	-	
	Male	1.603	0.840	3.061	0.153
Age (continuous)		1.032	0.986	1.079	0.175
Age	<70	1 (ref)	-	-	
	>70	2.697	0.830	8.768	0.099
Year	2017	1 (ref)	-	-	
	2018	0.555	0.107	2.869	0.483
	2019	0.903	0.230	3.554	0.884
	2020	0.843	0.157	4.539	0.842
Cohort	2017–2018	1 (ref)	-	-	
	2019–2020	1.089	0.348	3.412	0.884

Legend: CI = confidence interval; Sign = significativity.

**Table 3 jcm-12-04127-t003:** Quality of life in patients awaiting surgery (inguinal hernia).

	First Evaluation			Last Follow-Up			
	Median	IQR		Median	IQR		*p*-Value
Pain	3	0	5	2	0	5	0.813
Working activity limitation	0	0	3	0	0	4	0.746
Social activity limitation	1	0	3	0	0	3	0.348
Sport activity limitation	1	0	4	1	0	5	0.345

Legend: IQR = interquartile range.

## Data Availability

Data are available under request to the corresponding author.

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
