# Peer review of "Evaluating the Natural History of Groin Hernia from an “Unplanned” Watchful Waiting Strategy"

_jcm, 2023, doi:10.3390/jcm12124127_

Round 1

Reviewer 1 Report

1. In the present study, the differences between the baseline data (e.g. comorbidities) of the two groups were not well addressed. In other words, the baseline data collected were not comprehensive.

2. It's arguable that the data analysis of the femoral hernia was included.

3. The limitations were not fully explained.

4. Compared with previous studies, what are the novel and key perspectives? 

Author Response

  1. In the present study, the differences between the baseline data (e.g. comorbidities) of the two groups were not well addressed. In other words, the baseline data collected were not comprehensive.

Unfortunately data about comorbidities were not available and no considerations about could be made.

  1. It's arguable that the data analysis of the femoral hernia was included.

Dear reviewer; we perfectly agree that femoral and inguinal hernia are two different clinical problems with different outcomes and incidence of complications, as well described by our data. We decided to include also data about femoral hernia but with different and separate analysis from inguinal hernia.

  1. The limitations were not fully explained.

Limitation of the study are the retrospective nature of the study and the missing data. We improved the limitation section accordingly.

  1. Compared with previous studies, what are the novel and key perspectives? 

We think that our data do not provide key perspectives for new studies but confirm the evidences derived from well designed clinical trial performed in highly selected groups of patients, translating the evidences in a very heterogeneous, not selected cohort of patients that reflects the daily practice of a large general surgery unit.

Reviewer 2 Report

This paper addresses the issues regarding the risk of complications among patients with asymptomatic or mildly symptomatic groin hernia, exploiting the weird situation that occurred with the outburst of COVID pandemia, when elective surgery was canceled/post-poned and a watchful waiting strategy was imposed. The topic represents a kind of interesting experimental model, though quite different from a controlled clinical trial, due to the different setting. This has been clearly clarified in text by Authors and well represents the intrinsic limitations of this paper. On the other hand, from a real-life perspective, it represents a good observational longitudinal study. Conclusions are consistent with the available evidence and the rationale of the paper.

Author Response

thank you very much for your comments

Reviewer 3 Report

The authors performed an analysis on patients with inguinal and femoral hernias awaiting surgery and found that complication rates are low for inguinal hernia, and somewhat high for femoral hernias. Quality of life for inguinal hernia was not significantly modified during the waiting time.

The main strength of this article is the high number of patients in the study and the length to surgery is fairly high, showing rates of complication for asymptomatic hernias. The downside is that the article is retrospective, while existing Randomized Controlled Trials analysing watchful waiting already exist.

Q#1: In your opinion, what do you believe this article brings additionally to existing RCT?

The article is well written, abstract contains al necessary background information and is relevant to the article in question. The methods are adequately described. Results are presented concise, easy to understand. Discussions support the results section. Conclusions are short and concise.

The use of English in the article is adequate to an academic paper. Short sentences for ease of understanding are deployed. 

Q #2: Was EuraHS QoL used for patients with femoral hernia?

Q #3:  Patients were asymptomatic when initially screened? If so, have you assessed if they have remained asymptomatic up untill complication occured?

Kind regards

Author Response

Q#1: In your opinion, what do you believe this article brings additionally to existing RCT?

Despite the limit of a retrospective study our research focused on a real world experienced including all the patients evaluated in daily practice; the main difference is the absence of patients selection that in our opinion reinforces the results of our research. The discussion was improved accordingly.

Q #2: Was EuraHS QoL used for patients with femoral hernia?

The questionnaire was adopted in all patients; method section was updated for more clairity.

Q #3:  Patients were asymptomatic when initially screened? If so, have you assessed if they have remained asymptomatic up untill complication occured?

All patients selected for elective groin hernia repair were included in the study, without any selection based on symptoms. Our study reflects the daily practice of a large volume general surgery unit, providing evidences implementable in the real world, outside the highly selection of a clinical study.

The method section was updated for more clarity.

Reviewer 4 Report

This is a retrospective cross sectional cohort study to assess the impact of the epidemic on inguinal hernia patients. Fortunately, we did not find any significant adverse effects of waiting and the epidemic on patient surgery, providing us with a good reference. The vast majority of inguinal hernia patients do not undergo emergency surgery, I hope to see a table comparing the differences in surgical complications among all surgical patients from 2016 to 2018 compared to 2019 to 2021, which will make it more intuitive to determine that the epidemic has no impact on surgery and patient health.

Author Response

Thank you very much for your suggestion. The aim of our study was to evaluate the possible effect of delaying surgery on hernia complication (incarceration of strangulation). Data about complications after elective surgery are not consistent with the aim; moreover unfortunately we have not complete data on it.